# The Scale of Perceptions and Self-Participation in Hemodialysis: Development and Psychometric Evaluation

**DOI:** 10.3390/healthcare11233072

**Published:** 2023-11-30

**Authors:** Li-Yun Szu, Suh-Ing Hsieh, Whei-Mei Shih, Mei-Chu Tsai, Su-Mei Tseng

**Affiliations:** 1Department of Nursing, Taoyuan Chang Gung Memorial Hospital, Taoyuan City 33372, Taiwan; szu4069@cgmh.org.tw (L.-Y.S.); q22075@cgmh.org.tw (M.-C.T.); ts6852@cgmh.org.tw (S.-M.T.); 2Department of Nursing, Chang Gung University of Science and Technology, Taoyuan City 33302, Taiwan; 3Graduate Institute of Health Care, Chang Gung University of Science and Technology, Taoyuan City 33302, Taiwan; jeanshih168@frontier.com

**Keywords:** hemodialysis patient, scale of self-participation in hemodialysis, grounded theory

## Abstract

Hemodialysis patients undergo hemodialysis two to three times a week and must live together with the dialysis machine. The patient’s self-participation is to combine the patient’s own experience and professional knowledge to influence the care. A successful self-participation experience empowers patients to adapt to living with hemodialysis. However, few studies have been conducted regarding the subjective experiences of such patient participation. There is a lack of a self-participation dialysis life scale for hemodialysis patients. Therefore, this study aims to develop and evaluate a self-participation dialysis life scale for hemodialysis patients. The items for the self-participation dialysis life scale were confirmed through qualitative interviews based on grounded theory. After expert content validity evaluation, 435 hemodialysis patients were purposively sampled from hemodialysis centers in two regional teaching hospitals in Taiwan. Descriptive statistics, principal axis factoring, reliability analysis, Pearson’s correlation, and one-way ANOVA were used for data analysis. The results show that the item- and scale-content validity indices are 0.96 and 0.73, respectively. This scale is divided into two scales. The first part is “Scale of Perceptions of Hemodialysis”, including two factors. The overall can explain 66.34% of the cumulative variances. The second part is the “Scale of Self-Participation in Hemodialysis”, including four factors. The overall can explain 58.91% of the cumulative variances. The Cronbach’s α is 0.812 for “Perceptions of Hemodialysis” and 0.959 for “Self-Participation in Hemodialysis”, respectively. The self-participation dialysis life scale has good reliability and validity and can be used to evaluate the implementation of a patient’s self-participation in their hemodialysis life. Discussing or consulting with patients based on their characteristics, life priorities, and their desired life on dialysis is essential for a high-quality of clinical care among hemodialysis patients.

## 1. Introduction

In 2019, 134,608 individuals were newly diagnosed with end-stage renal disease (ESRD), representing an increase of 2.7% from the previous year; 85% (492,096 people) of those with incident end-stage renal failure (ESRD) initiated in-center hemodialysis (HD), up 1.7% from 2018 and 34.5% from 2009 [1]. For patients with ESRD, hemodialysis is a priority and a quick life-saving approach. Because it cannot completely replace kidney function, many uncomfortable symptoms or side effects occur after hemodialysis. Flythe et al.’s (2015) systematic literature review pointed out that 23 symptoms may occur after hemodialysis [2]. In addition, there are two to three weekly hemodialysis sessions, each takes 3–4 h. Psychological problems such as role restrictions or social isolation happen after dialysis, affecting everyday life [3,4,5,6,7]. The five-year survival rate of hemodialysis patients in Taiwan is 54.3%, and the ten-year survival rate is 33.8% [8]. The study participants’ average number of years of hemodialysis is 9.10 ± 7.37 years. During their long lives, they will experience many side effects of hemodialysis, spend time in various outpatient clinics, and be repeatedly hospitalized [8]. The experience of helplessness is associated with different disease progressions, and it affects the quality of life [7]. Hence the Executive Order on Advancing American Kidney Health, which was released in July 2019 (U.S. Department of Health and Human Services, 2019) and sets forth a target for the utilization of home dialysis and preemptive transplantation that is much higher than HD or peritoneal dialysis (PD) in the U.S. [1]; however, more healthcare support is needed. The enabling process can facilitate patient empowerment by providing knowledge tools, techniques, support, and self-management interventions [9,10]. Castro (2016) indicated that “patient-centeredness is based on mutually beneficial partnerships between the patient, his family, and the healthcare provider, and is characterized by open communication of knowledge [9]. There is an exchange of experiential knowledge and clinical knowledge” [9]. Patients are increasingly invited to take an active role in in their care at more strategic levels, such as shared decision making in medicine and healthcare.

The gap between patients’ experiential knowledge and professional knowledge then requires patients to participate in the decision-making process for hemodialysis patients [9,11]. Patients are regarded as experts on their physical symptoms and discussions and consultations should take place with each patient based on their characteristics and life priorities. The patient’s knowledge, experience, and learning should be internalized to the behavioral level [12], rather than using the medical profession to train patients to adapt to hemodialysis life. Nevertheless, using past care experience to persuade patients to make various life changes recommended by medical staff is important [10], but this passive acceptance of professional care information from medical staff cannot meet the patient’s health needs [13]. Hence, self-participation is a critical patient-centered care strategy. Then, patients can genuinely participate in hemodialysis life. Nursing staff can promote patient participation. The care must be adapted to the risks and characteristics of each patient, and good assessment skills and tools must be available to enable nursing staff to play a critical role in patient self-participation [14]. Arestedt et al. (2019) and Szu et al. (2021) illustrate the importance of hemodialysis patients’ self-involvement in dialysis life [15,16]. Patients’ self-participation in hemodialysis life can bridge the gap between patients’ experiential knowledge and professional knowledge. Therefore, this study aims to use the qualitative research method with grounded theory to identify empirical indicators for “the Scale of Perceptions of and Self-Participation in Hemodialysis (SPSPHD)” and examine the scale’s reliability and validity through the quantitative research method.

## 2. Materials and Methods

### 2.1. Design

This study was a psychometric test using a structural questionnaire to conduct a cross-sectional survey that is part of a large study [17].

### 2.2. Participants and Setting

Purposive sampling was used for this study for hemodialysis patients in the hemodialysis centers of two regional teaching hospitals in northern Taiwan. The inclusion criteria included patients with end-stage renal disease diagnosed by a doctor, who have continued regular hemodialysis for more than one year, have normal hearing or use hearing aids to assist hearing, can communicate in Mandarin and/or Taiwanese, and agree to participate in this study. The exclusion conditions excluded patients with self-reported unstable medical conditions, such as cancer, severe cardiopulmonary, liver, and kidney dysfunctions, critical illness patients, and hemodialysis patients with mental disorders. The sample size was estimated to require 10–15 people per question [18]. In this study, 450 questionnaires were distributed and 439 were retrieved; four invalid questionnaires were excluded. The response rate of valid questionnaires was 97.67%.

### 2.3. Instruments

The study tools included sociodemographic and clinical characteristics, the Hemodialysis Patient Self-Management Scale, and the Scale of Perceptions of and Self-Participation in Hemodialysis Dialysis. First, the sociodemographic and clinical characteristics were structured in self-report questionnaires; the information collected included age, gender, marital status, education level, religion, work status, self-assessed health status, comorbidities, number of years of continuous hemodialysis, and companions during hemodialysis.

Secondly, the Hemodialysis Self-Management Instrument (HD-SMI), a hemodialysis patient self-management behavior scale constructed by Song and Lin (2009) [19] based on Curtin et al. (2004) [20], includes most of the items for assessing the self-care activities of dialysis patients during treatment but lacks measurement of executing self-care activities in daily life [20]. Moreover, with the differences in medical culture domestically and abroad, a self-management scale for hemodialysis patients was developed and its reliability and validity were tested [19]. The CVI (Content Validity Index) of the first version of the scale was 0.92, and the expert content validity CVI of the second round was 0.92. The direct oblimin rotation (*n* = 196) results of exploratory principal component analysis showed that four factors could explain 45.13% of the total variation. The scale has 20 questions and a 4-point Likert scale was applied, with one indicating never and five indicating always. The possible total score range is between 20 and 100. The higher the score, the better the self-management behavior. The internal consistency of Cronbach’s α of the total scale is 0.87. The Cronbach’s α of the four factors ranges from 0.70 to 0.78, and the correlation coefficient of test–retest reliability after 2 weeks is 0.86 (*p* < 0.001). This indicates that the scale has good reliability. The internal consistency Cronbach’s α of the total scale for the study group was 0.93.

Finally, for the Scale of Perceptions of and Self-Participation in Hemodialysis (SPSPHD), due to the lack of domestic measurement tools suitable for hemodialysis patients’ self-participation in dialysis life, qualitative research was conducted with the grounded theory research method in the beginning for the comparison and analysis of hemodialysis patients’ self-involvement in hemodialysis life experience and, based on the results of the qualitative data analysis, a 42-item original scale was constructed. The eligible participants were recruited from an outpatient hemodialysis clinic with 126 beds at a medical center in Taiwan using purposive sampling. The well-adapted patients were referred by attending physicians or the head nurse [16]. Six experts and one experienced hemodialysis patient were invited to evaluate the content validity of the 42-item original SPSPHD, including two academic researchers, two clinical renal nurses, a head nurse, a director of the nephrology department, and a social media manager for dialysis life on Facebook. A 4-point Likert scale was utilized, with one indicating not relevant and four indicating very relevant to appraise the content validity of the 42-item original SPSPHD. This tool was modified and the questions merged based on comments from experts. The second version of the scale contained a total of 37 questions, including 8 questions in the first part, to measure the positive and negative feelings about hemodialysis within the past three months. For the three questions regarding negative emotions, reverse scoring was used with a 5-point Likert scale (4 = Always: feeling this way almost every day; 3 = Often: feeling this way more than three days a week (inclusive) every month; 2 = Sometimes: feeling this way two days or less a week every month; 1 = Occasionally: feeling this way two days or less every month; 0 = Rarely: I never felt this way in the past three months. The possible total score range is between 0 and 32. The higher the score the greater the feeling related to hemodialysis. In the second part, 30 questions were used to measure the level of self-participation in hemodialysis life during the past 3 months; 29 questions were structured options and 1 question was an open question. The scoring standard was a 5-point Likert Scale: 0 = Almost entirely impossible: unable to do it or can only complete 10–20% (0–20%); 1 = Occasionally can do it: only 21–49% can be done; 2 = Often: can only complete half (50–69%); 3 = Mostly: can achieve 70–89%; and 4 = Almost entirely: more than 90% were complete (90–100%). The possible total score ranges from 0 to 116. A higher score means a higher level of self-engagement in hemodialysis.

### 2.4. Procedure

The research procedure is shown in Figure 1. After IRB approval, from August 2018 to October 2018 patients who had undergone outpatient hemodialysis for 1 year, were conscious, and were over 20 years old were recruited after the principal investigator explained the purpose and process of the research in the family lounge of the dialysis outpatient. The patients who agreed to participate were interviewed face-to-face and completed the questionnaire within 24 h. The interviews were analyzed to summarize the themes and a framework. The qualitative data were analyzed. A 42-item original scale was developed based on the qualitative data analysis results and was evaluated for expert content validity. The items were modified and merged based on expert opinions and the second version of the scale was developed. Patients were invited for a qualitative interview again to fill in the questionnaire and provide feedback on the scale. From 20 April 2019 to 19 April 2021, through introductions by nephrologists or hemodialysis room patients, participants who met the inclusion criteria were screened. Afterward, the research assistant explained the purpose and process of the study to the participants and obtained their consent before filling in the questionnaire. For some older and illiterate participants, the research assistant asked them the questions and later filled in the answers. After the questionnaire was completed, the completeness of the questionnaire content was reviewed immediately and entered into the Excel database gradually. Finally, reliability analysis, criterion-related validity, and constructive validity analyses were conducted.

### 2.5. Ethical Considerations

This study was approved by the human trial institutions (No.201900456B0 and 20180304R) of two teaching hospitals in northern Taiwan. Before the questionnaire was completed, it was explained to participants that they could withdraw from the study without affecting their entitlement to treatment. In addition to the subject consent form, the questionnaire was replaced by a code to ensure the participant’s anonymity.

### 2.6. Data Analysis

The data were analyzed using SPSS Statistics for Windows 22 (IBM, Armonk, NY, USA). Normality, linearity, outliers, multicollinearity of assumptions for factor analysis, Pearson’s correlation, and one-way ANOVA were checked. Descriptive statistics of the frequency, percent, average, and standard deviation were used for participants’ sociodemographic and clinical characteristics. The internal consistency of the Scale of Perceptions of and Self-Participation in Hemodialysis Dialysis was analyzed using reliability analysis. Pearson’s correlation was used to assess the concurrent validity of the criteria-related validity between the HD-SMI and the SPSPHD. Principal axis factoring (PAF) of the exploratory factor analysis with Promax rotation was used to evaluate the construct validity based on initial eigenvalues (≥1), factor loadings (>0.35), and scree plots.

## 3. Results

### 3.1. Sociodemographic and Clinical Characteristics of the Participants

The participants were aged between 34 and 91 years old, with an average of 63.15 ± 11.62 years old. The majority were men (52.4%), most were married (74.9%), followed by unmarried (11.5%). The education level was mainly up to elementary school (32.6%), followed by high school/higher vocational (30.3%). Most were religious (84.6%), with more Buddhist (47.1%) than Taoist (31.5%). Eighty percent of the participants were unemployed (80.9%), more than 60% lived with their spouse (64.8%), and only 2.3% lived alone. A total of 84.1% of the participants had a comorbidity, including hypertension (64.8%), followed by diabetes mellitus (40.2%); a total of 37.0% had one comorbidity. The average number of hemodialysis years was 9.10 ± 7.37 years, with three hemodialysis sessions a week being the most common (94.9%), followed by hemodialysis twice a week (5.1%). Transportation to and from the hemodialysis room was mostly by car (39.8%), followed by bus (24.6%) or self-owned vehicle (18.2%). The majority (55.4%) were accompanied during hemodialysis by family members (49.4%), followed by caregivers (4.4%)

### 3.2. Content Validity

The Item-Content Validity Index (I-CVI) and Scale-CVI (S-CVI) of the SPSPHD were 0.96 and 0.73, respectively.

### 3.3. Construct Validity—Principal Axis Factoring of the Exploratory Factor Analysis

Exploratory factor analysis (EFA) was conducted on the original 37 questions SPSPHD (8 questions on perception and 29 questions on self-participation) for PAF. The results showed that this scale was divided into two scales so that the PAF was separated. The first part was the “Scale of Perceptions of Hemodialysis”, and the Kaiser–Meyer–Olkin (KMO) measurement sample size suitability (a measure of sampling equality) was 0.783. The measure of sampling equality (MSA) between individual items ranged from 0.677 to 0.882, and the Bartlett test of sphericity showed significant differences (*p* < 0.001), with initial eigenvalue values of 3.712 and 2.264, respectively. This section is divided into two factors. The first factor can explain 42.48% of the variance, including questions four to eight (five questions), named atmosphere during hemodialysis (Table 1). The second factor can explain 23.86% of the variance, including questions one to three (three questions), named negative emotions during hemodialysis. The overall interpretability of both factors is 66.34%, with a cumulative variance of variance (Table 2). The scree plot is presented in Figure 2.

The second part is the “Scale of Self-Participation in Hemodialysis”; the Kaiser–Meyer–Olkin (KMO) for measuring the sample size was 0.952 and the MSA between the individual items ranged from 0.897 to 0.974. The Bartlett test of sphericity showed significance (*p* < 0.001), and the initial eigenvalues were 13.81, 2.45, 1.32, and 1.04, respectively. Divided into four factors, the first factor explained 46.28% of variance, including questions 21, 22, 24, 26, 27, 28, and 29 (seven questions), named “Creating a New Life”. The second factor explained 7.22% of variance, including questions 6, 10, 12, 13, 14, 15, 16, 17, and 18 (nine questions), named implementing self-care. The third factor explained 3.09% of the variance, including questions 1, 2, 3, 4, 5, 7, 8, 9, and 11 (nine questions), named “Adjusting and Facing Life of Hemodialysis “. The fourth factor explained 2.32% of variance, including questions 19, 20, 23, and 25 (four questions), named active sharing and shared decision (Table 3). The four factors as a whole explained 58.91% of the cumulative variance (Table 2). The scree plot is presented in Figure 3.

### 3.4. Construct Validity—Known Groups Method

One-way ANOVA analysis revealed a significant difference in hemodialysis patients’ average perceptions of hematology scores (137 vs. 298) in two different teaching hospitals in different regions (26.54 ± 5.33 vs. 24.19 ± 7.07, *p* = 0.001). Additionally, there is a significant difference in the average scores for self-participation in hemodialysis (86.49 ± 22.38 vs. 78.32 ± 23.15, *p* = 0.001).

### 3.5. Concurrent Validity of the Criterion-Related Validity

Pearson’s correlation analysis between the three metrics “Scale of Perceptions of Hemodialysis”, “Scale of Self-Participation in Hemodialysis”, and “HD-SMI” showed a significant mild positive correlation between “Scale of Perceptions of Hemodialysis” and “Scale of Self-Participation in Hemodialysis” (*r* = 0.37, *p* < 0.001), a significant mild positive correlation (*r* = 0.26, *p* < 0.01) between “Scale of Perceptions of Hemodialysis” and “HD-SMI”, and a significant moderate positive correlation between “Scale of Self-Participation in Hemodialysis” and “HD-SMI” (Table 4).

### 3.6. Internal Consistency Reliability

The SPSPSPHD is divided into two parts. Firstly, the Scale of Perceptions of Hemodialysis’ Cronbach’s α is 0.812. The two scales are the atmosphere during hemodialysis and negative emotions during hemodialysis, which have Cronbach’s α values of 0.910 and 0.836, respectively, with a corrected item-total correlation between 0.650 and 0.826 (Table 1). Secondly, the Cronbach’s α of the Scale of Self-Participation in Hemodialysis is 0.959. The Cronbach’s α for four factors is 0.927 for creating a new life, 0.902 for implementation of self-care, 0.893 for adjusting and facing the hemodialysis life, and 0.858 for active sharing and shared decisions. The corrected item-total correlation is between 0.454 and 0.886 (Table 3).

In Table 4, the correlation between the two parts of the SPSPHD scale is presented; there is no significant negative correlation between the two factors of the Scale of Perceptions of Hemodialysis (*r* = −0.04). The correlation coefficient between the atmosphere of hemodialysis and the four factors of the Scale of Self-Participation in Hemodialysis was between −0.18 and 0.29 and there was a significant weak negative correlation (*p* < 0.01). The correlation coefficient between negative emotions of hemodialysis and the four factors of the Scale of Self-participation in Hemodialysis ranged from 0.20 to 0.38, with a significant weak positive correlation (*p* < 0.01), whereas the correlation coefficient between the four factors of the Scale of Self-participation in Hemodialysis ranged from 0.59 to 0.80, with a significant moderate to highly positive correlation (*p* < 0.01). Among these factors, the highest correlation coefficient is between the implementation of self-care and adjusting and facing hemodialysis life (*r* = 0.80) and the weakest correlation coefficient is between adjusting and facing hemodialysis life and active sharing and shared decisions (*r* = 0.59).

### 3.7. Descriptive Statistics of the SPSPHD

Table 5 shows that the two-part SPSPHD presents a grand mean and standard deviation of 5.21 ± 6.05 and 10.14 ± 2.56 for two factors, respectively, and a grand mean and standard deviation of 18.72 ± 7.27, 27.00 ± 7.37, 24.64 ± 7.70, and 10.53 ± 4.16 for four factors. The mean and standard deviation (SD) of each factor is divided by the total mean and by the number of questions in descending order. Factor 2 (the negative emotions during hemodialysis): 3.38 ± 0.33, factor 2 (implementation of self-care): 3.00 ± 0.82, factor 3 (adjusting and facing hemodialysis life): 2.74 ± 0.86, factor 1 (creating a new life): 2.67 ± 1.04, factor 4 (active sharing and shared decisions): 2.63 ± 1.04, and factor 1 (the atmosphere during hemodialysis): 1.04 ± 1.21.

## 4. Discussion

### 4.1. The Aim of the Study and Content Validity

This study develops and evaluates the expert content validity evaluation, construct validity, and internal consistency reliability of the SPSPHD. The I-CVI and S-CV are 0.96 and 0.73, respectively, indicating each item’s high expert content validity and the overall scale.

### 4.2. Construct Validity—PAF of the EFA

This study aims to find common factors among many observed variables. The error in the estimation process is small and the accuracy of the factor loading is high. Therefore, PAF was used to explain the validity. Additionally, the correlation coefficients between most items were >0.30, so Promax rotation was used. EFA uses PAF and Promax rotation [18]. Two scales are used to divide the SPSPHD. The first scale is “Perceptions of Hemodialysis”. The KMO value is 0.783, whereas the MSA between individual items is from 0.677 to 0.882. Bartlett’s test of sphericity showed significance (*p* < 0.001). The first factor explains 42.48% of the variance, the second factor explains 23.86% of the variance, and both factors explain 66.34% of the cumulative variance (Table 2). The second scale is “Self-Participation in Hemodialysis”. The KMO value is 0.952, whereas the MSA between individual items is from 0.897 to 0.974. Bartlett’s test of sphericity shows significance (*p* < 0.001). The scale is divided into four factors. The four factors explain 2.32–46.28% of the variance. All four factors together explain 58.91% of the cumulative variance. This indicates that the SPSPHD has construct validity.

“Atmosphere during hemodialysis” and “Negative emotions during hemodialysis” were antecedent themes based on patients’ perceptions of hemodialysis and the primary precursor factor for self-participation [16]. Upon learning that they required HD, patients experienced negative feelings arising from their need to depend on others and the discrimination that they would face. HD patients may develop a sense of worthlessness or depression. These negative feelings may be reduced through joining a group in which they no longer fight the disease alone and may work together with people with the same condition and treatment.

“Creating a New Life”, “Implementing Self Care”, “Adjusting and Facing Hemodialysis”, and “Actively Sharing and Making Joint Decisions” were four interactive themes. First, “creating a new life” means that hemodialysis patients can discover the joys of life such as developing interests, finding the fun, generating positive energy, attending outdoor and social activities, and solving negative feelings. Second, ”implementation of self-care” represents that hemodialysis patients can perform self-care business such as checking bleeding and infectious signs on acupuncture sites, checking blood flow vibration, preparing supplies, food, and recreation, selecting adequate medical care strategies for sick conditions, accepting advice or help from others, and returning to life and/or work. Third, “adjusting and facing the dialysis life” displays that hemodialysis patients can take action and overcome the restrictions of the illness and treatment such as water intake, diet, medicine, exercise, and side effects. Fourth, “active sharing and shared decisions” indicates that hemodialysis patients can share their experiences of hemodialysis life with others, seek assistance from others, help other hemodialysis patients, discuss issues with medical personnel, and make decisions on treatment.

### 4.3. Construct Validity—Known Groups Method

The known groups method shows that the average scores on the Scale of Perceptions of Hemodialysis for hemodialysis patients in the hemodialysis center of the teaching hospital in region A were significantly higher than the hemodialysis patients in the hemodialysis center of the teaching hospital in region B (26.54 vs. 24.19). Additionally, the average scores for Self-Participation in Hemodialysis were significantly higher than those of hemodialysis patients in the hemodialysis center of the teaching hospital in region B (86.49 vs. 78.32). This means that the hemodialysis patients in the hemodialysis center of the teaching hospital in region A have a better perception of hemodialysis and self-participation in hemodialysis life. This might be because the hemodialysis patients served by the hemodialysis center of the teaching hospital in region A have higher socioeconomic status and reside in a metropolitan area. In addition, the medical staff of the hemodialysis center provide more patient-centered medical services. It also indicates that the SPSPHD has construct validity.

### 4.4. Concurrent Validity of the Criterion-Related Validity

Among the “Perceptions of Hemodialysis”, “Self-Participation in Hemodialysis”, and “HD-SMI” constructed in this study, the Pearson’s correlation shows that there is a significant slight positive correlation between the “Perceptions of Hemodialysis” and “Self-Participation in Hemodialysis” (*p* < 0.001), there is a significant small positive correlation between the “Perceptions of Hemodialysis” and “HD-SMI” (*r* = 0.262, *p* < 0.001), and there is a significant moderate positive correlation between the “Self-Participation in Hemodialysis” and “HD-SMI” (*r* = 0.590, *p* < 0.001). These results indicate that this tool has concurrent validity of the criterion-related validity.

### 4.5. Internal Consistency Reliability

A Cronbach’s α ≥ 0.80 indicates good internal consistency reliability. The Cronbach’s α of “the SPSPHD” total scale is 0.950. The scale is divided into two parts: “Perceptions of Hemodialysis” (Cronbach’s α 0.812) and “Self-Participation in Hemodialysis” (Cronbach’s α 0.959). The former is divided into two factors, where Cronbach’s α is 0.836–0.910. The latter is divided into four factors, where Cronbach’s α is 0.858–0.927. In terms of the correlation between the two parts of the SPSPHD scale, there is no significant negative correlation between the two factors of the “Scale of Perceptions of Hemodialysis ” (*r* = −0.04). There is a significant weak negative correlation between the “Atmosphere during hemodialysis” and the “Scale of Self-Participation in Hemodialysis” (*p* < 0.01). There is a significant weak positive correlation between the “Negative emotions during hemodialysis” and the “Scale of Self-Participation in Hemodialysis” (*p* < 0.01). There is a significant moderate-to-high positive correlation among the “Scale of Self-Participation in Hemodialysis (*p* < 0.01), among which the highest correlation is between “Implementing self-care” and “Adjusting and Facing the Hemodialysis Life” (*r* = 0.80) and the weakest correlation is between “Adjusting and Facing the Hemodialysis Life” and “Proactively sharing and sharing decisions” (*r* = 0.59). The weakest point for dialysis patients is “Active Sharing and Shared Decisions”, which may be related to the age group (average 63.15 ± 11.62 years old) and the education level (84.7% have high school vocational education or below) of the participants or because they lack self-confidence or the ability to share personal feelings. The best point is “Implementing self-care” during the hemodialysis Life, which shows that when patients participate in hemodialysis life and face their symptoms after hemodialysis they can take care of themselves. This also proves what the scholars Mok and Martinson (2000) suggested “Patients are experts on physical symptoms” [11]. After discussing and consulting with each patient based on their characteristics and life priorities, they should internalize knowledge, experience, and learning into their behavioral levels instead of using health education to train patients.

### 4.6. Descriptive Statistics of the SPSPHD

The highest mean is the negative emotion of hemodialysis in factor 2 (3.38), and the lowest is the atmosphere of hemodialysis in factor 1 (1.04). This may be because in the second part of the scale the patient’s intention to participate in hemodialysis life and negative emotions for hemodialysis have the greatest impact and the atmosphere in the hemodialysis room has less impact, which is also a manifestation of patient centeredness. This shows that nursing staff should pay attention to the psychological level of hemodialysis patients. If negative emotions are detected, they are encouraged to express negative feelings promptly and timely referrals should be made to psychological treatment, which can increase patients’ willingness to participate in dialysis life.

This study found that the value of survival and living and integrating into the friendly atmosphere of the dialysis room are the pioneer categories for patients to self-participate in hemodialysis life. They are also the driving forces for motivating self-participation in hemodialysis life and should be actively valued. Most participants have religious beliefs (84.6%). It is recommended that medical staff use the power of love (family or key person) to increase the value of survival and reduce the sense of hopelessness, especially for those over 65 years old, through life review and to reduce the burden on family members. In terms of burdening or relying on others to guide oneself in hemodialysis life, Santo et al. (2017) discovered a significant positive correlation between positive religious/spiritual adjustment scores and general health and vitality [21]. In recent years, many research results have shown that religious or spiritual interventions in chronic disease patients can help trigger positive emotional responses to stress. Religion and spirituality may affect hemodialysis patients’ overall quality of life index, including their commitment and adherence to hemodialysis treatment [22]. It is suggested that meditation and religious or spiritual interventions can help with positive thinking and emotional stability in coping with hemodialysis for inducing love, aliveness (perceptions of hemodialysis), and promoting self-participation in hemodialysis.

The higher SPSPHD scores indicate active participation in hemodialysis life and the author [16] has shown in the factors related to quality of life of hemodialysis patients during the COVID-19 pandemic study that there is a significantly positive correlation between the self-involvement and quality of life of hemodialysis patients.

In some cases, arrangements were made to seat HD patients with a successful self-participation by tracking records for patients undergoing their first HD session or patients who had not adapted well to the process. Finally, the SPSPHD can be applied to future research for assessing patients with peritoneal dialysis and chronic kidney diseases, which both require self-care capabilities and demonstrate the importance of patient self-participation.

### 4.7. Limitation of the Study

There are some limitations in this study. Firstly, the study only focuses on patients from the outpatient hemodialysis centers of two teaching hospitals in the North District. In the qualitative research stage theoretical saturation sampling was used and in the scale development stage purposive sampling is used to recruit research subjects. The representativeness of the samples is insufficient, so the research results cannot be applied to hemodialysis cases in other regions of Taiwan. Secondly, the “SPSPHD” developed in this study has good content validity, concurrent validity, and construct validity. However, due to time, manpower, and regional limitations, a large number of samples were not collected for reliability and validity testing. In the future, we will continue to conduct reliability and validity testing (e.g., confirmatory factor analysis) on the developed scale for more significant sample research subjects, making it a more rigorous and reliable measurement tool.

## 5. Conclusions

The Scale of Perceptions of and Self-Participation in Hemodialysis for hemodialysis patients constructed and developed by this research institute was validated for validity and reliability, indicating that it can be used as an evaluation tool for patients’ self-participation in hemodialysis life execution. In the self-participation hemodialysis life of hemodialysis patients, in addition to hemodialysis techniques, nursing staff should also create a friendly atmosphere in the hemodialysis room, allow hemodialysis patients who share common experiences to fully express the pain of suffering from the disease, and have a sense of belonging when experiencing hardships. Nursing staff should empathize and praise patients for their bravery in receiving injections, encourage the patient to overcome fatigue, encourage the patient to express negative or sad emotions, accompany the patient, pay close attention to patients to identify whether they have the idea of giving up treatment or giving up survival, identify the influence of the patient’s key person (close relationship person), focus on patients, and truly become one of the support systems for hemodialysis patients. The SPSPHD is constructed based on patient-centered care, discussing or consulting with patients based on their characteristics and life priorities, and the patient’s desired life on hemodialysis. Only when patients truly participate in hemodialysis life and integrate into hemodialysis life is their SPSPHD score higher, their self-care and self-management better (positively correlated), and their quality of care, HD quality of life, and happiness improved. Therefore, medical teams must scrutinize the guidance content for hemodialysis patients more carefully instead of imposing professional knowledge on patients and forcing patients to comply. Hemodialysis patients can better understand the importance of self-participation in hemodialysis life, which will improve their quality of life.

## Figures and Tables

**Figure 1 healthcare-11-03072-f001:**
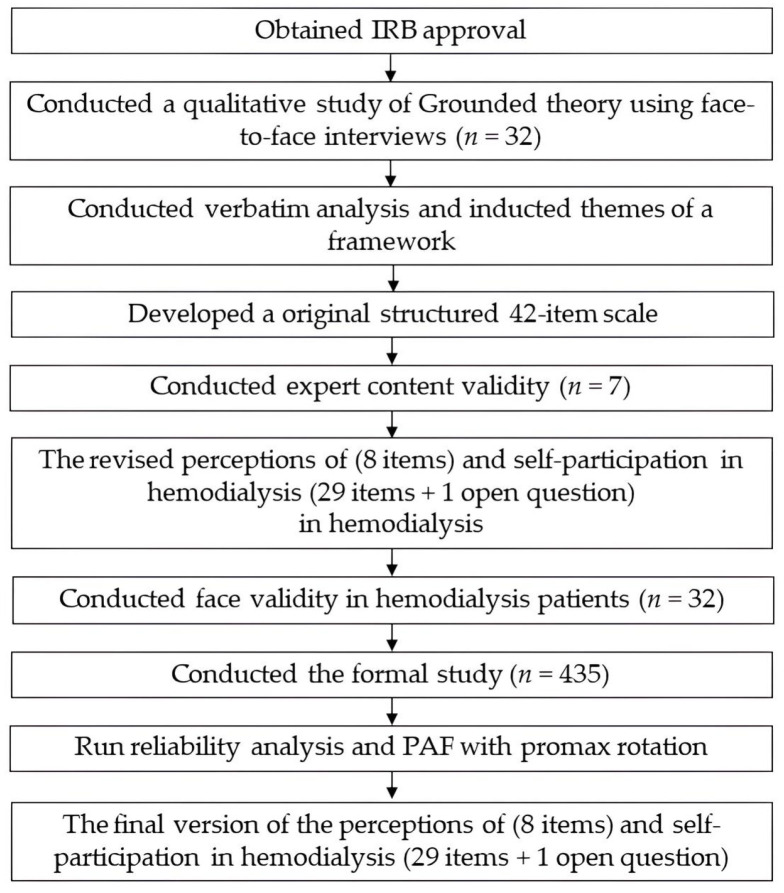
Procedure of this study.

**Figure 2 healthcare-11-03072-f002:**
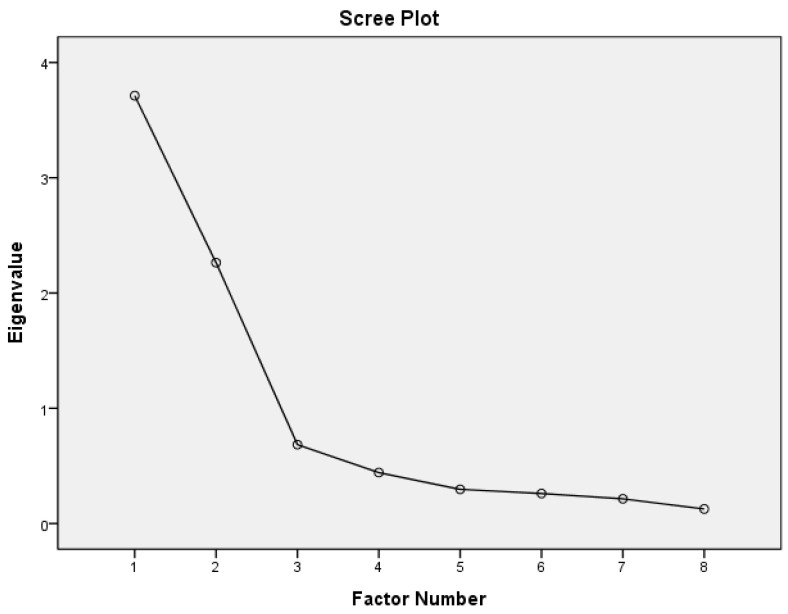
Scree plot of the Scale of Perceptions of Hemodialysis.

**Figure 3 healthcare-11-03072-f003:**
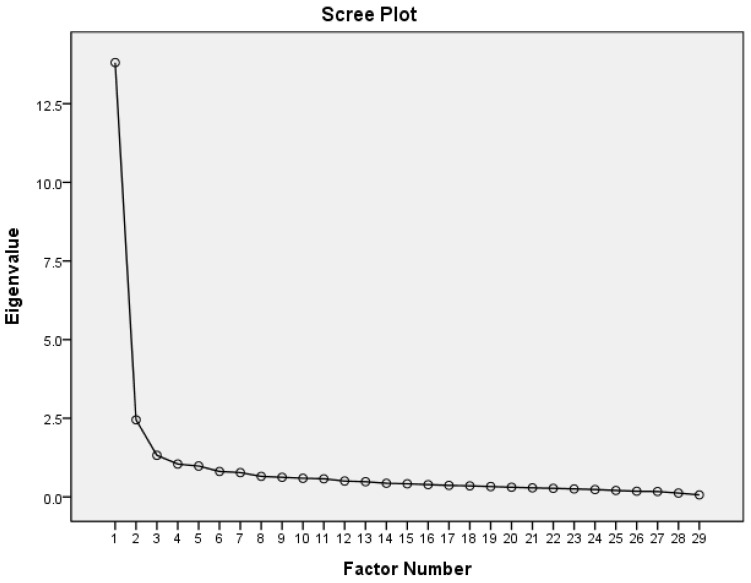
Scree plot of the Scale of Self-Participation in Hemodialysis.

**Table 1 healthcare-11-03072-t001:** Factors and items of the Scale of Perceptions of Hemodialysis using PAF with Promax rotation (*n* = 435).

Factors and Items (Cronbach’s α)	Pattern Matrix	Structure Matrix	Corrected Item-Total Correlation
Perceptions of Hemodialysis (α = 0.812)			
Factor 1: Atmosphere during hemodialysis (α = 0.910)			
6. The friendly dialysis environment makes me feel warm inside.	0.882	0.882	0.826
7. I have a pleasant time with my hemodialysis partners.	0.865	0.863	0.801
8. I will refer to the experience of hemodialysis partners and learn about the life of self-care and dialysis.	0.829	0.829	0.770
5. For my unfulfilled wish, I will continue to receive hemodialysis.	0.778	0.778	0.749
4. In order to love my loved relatives or friends, I will continue to undergo hemodialysis.	0.756	0.758	0.732
Factor 2: Negative emotions during hemodialysis (α = 0.836)			
2. I need to rely on others to become a burden on my family and feel worthless.	0.842	0.843	0.735
1. I feel sad and worthless.	0.826	0.826	0.723
3. I feel discriminated against and feel worthless.	0.719	0.719	0.650

**Table 2 healthcare-11-03072-t002:** Total variance explained by 2 parts of the SPSPHD (*n* = 435).

	Initial Eigenvalues			Extraction Sums of Squared Loadings		
Factor	Total	% of Variance	Cumulative %	Total	% of Variance	Cumulative %
Scale of Perceptions of Hemodialysis						
Factor 1: Atmosphere during hemodialysis	3.71	46.40	46.40	3.40	42.48	42.48
Factor 2: Negative emotions during hemodialysis	2.26	28.30	74.70	1.91	23.86	66.34
Scale of Self-Participation in Hemodialysis						
Factor 1: Creating a new life	13.81	47.61	47.61	13.42	46.28	46.28
Factor 2: Implementation of self-care	2.45	8.45	56.06	2.09	7.22	53.50
Factor 3: Adjusting and facing hemodialysis	1.32	4.56	60.62	0.90	3.09	56.59
Factor 4: Active sharing and shared decisions	1.04	3.60	64.22	0.67	2.32	58.91

**Table 3 healthcare-11-03072-t003:** Factors and items on the Scale of Self-Participation in Hemodialysis using principal axis factoring with Promax rotation (*n* = 435).

Factors and Items (Cronbach’s α)	Pattern Matrix	Structure Matrix	Corrected Item-Total Correlation
Self-Participation in Hemodialysis (α = 0.959)			
Factor 1: Creating a new life (α = 0.927)			
28. I can develop interests in life.	0.976	0.948	0.886
27. I can find the fun in my life.	0.957	0.932	0.878
29. I can generate positive energy.	0.853	0.902	0.852
22. I can actively attend outdoor activities or social events to adjust the dialysis schedule.	0.543	0.734	0.733
24. I can find fun in my life or through outdoor activities.	0.543	0.774	0.768
26. I can arrange my transportation to and from the dialysis unit.	0.498	0.606	0.593
21. I can handle the negative emotions caused by dialysis.	0.375	0.704	0.697
Factor 2: Implementation of self-care (α = 0.902)			
16. I can check the bleeding (bruise) at the blood vessel and the infectious signs (for example redness, swelling, hotness, pain, etc.) at the injection site after hemodialysis.	0.939	0.850	0.776
17. I can feel the care and respect for the medical staff.	0.769	0.754	0.698
14. I can choose a suitable medical care strategy when I feel sick.	0.731	0.836	0.799
12. I can check for blood flow vibration or sizzling sound in the hemodialysis vessels.	0.723	0.697	0.632
13. I am confident that I can deal with physical discomfort after hemodialysis, such as palpitations, dizziness, cramps, itchy skin, etc.	0.621	0.786	0.771
18. I gradually returned to the living conditions and work I used to have before hemodialysis.	0.426	0.610	0.609
6. I can deal with the pain caused by needles in my way.	0.373	0.634	0.605
10. I can prepare the supplies needed and food and recreation for myself before hemodialysis.	0.369	0.637	0.622
15. I can accept advice or assistance from other hemodialysis patients to adjust my hemodialysis life.	0.352	0.629	0.617
Factor 3: Adjusting and facing the dialysis life (α = 0.893)			
7. When I don’t want to exercise because busy or tired, I can still exercise because of hemodialysis.	0.814	0.679	0.610
8. I can adjust the water intake and diet based on the amount of fluid removal each time.	0.742	0.817	0.758
5. I can force myself to do the things that are beneficial for hemodialysis.	0.725	0.682	0.634
9. I can try various methods to satisfy the desire for water intake and a normal diet.	0.650	0.786	0.728
2. I can adjust my diet and medicine at home and social events based on the blood laboratory report.	0.632	0.750	0.719
3. I can adjust to the original lifestyle due to the hemodialysis life.	0.540	0.754	0.730
4. I can seek spiritual sustenance such as religious beliefs and good relationships, and I can face hemodialysis positively.	0.457	0.653	0.629
1. I can encourage myself to handle the life changes after hemodialysis.	0.433	0.695	0.673
11. I can pay attention to the setting and blood pressure data on the hemodialysis machine.	0.379	0.475	0.454
Factor 4: Active sharing and shared decisions (α = 0.858)			
20. I treat medical staff or relatives and friends as people to talk to about life and disease-related issues.	0.772	0.787	0.728
25. I can actively seek help from others.	0.738	0.817	0.729
23. I can discuss with the medical staff or decide the treatment I’m expecting.	0.698	0.783	0.711
19. I’m willing to share the experience of hemodialysis life and help other hemodialysis patients.	0.506	0.707	0.647

**Table 4 healthcare-11-03072-t004:** Pearson’s correlation of the two and four factors of the SPSPHD (*n* = 435).

Factor	F1	F2	F1	F2	F3	F4
Scale of Perceptions of Hemodialysis						
Factor 1: The atmosphere during hemodialysis	1.00					
Factor 2: The negative emotions during hemodialysis	−0.04	1.00				
Scale of Self-Participation in Hemodialysis						
Factor 1: Creating a new life	−0.18 **	0.38 **	1.00			
Factor 2: Implementation of self-care	−0.29 **	0.27 **	0.64 **	1.00		
Factor 3: Adjusting and facing hemodialysis life	−0.24 **	0.25 **	0.64 **	0.80 **	1.00	
Factor 4: Active sharing and shared decisions	−0.27 **	0.20 **	0.77 **	0.65 **	0.59 **	1.00

Note. ** Pearson’s correlation at the *p* < 0.01 level (2-tailed).

**Table 5 healthcare-11-03072-t005:** Descriptive distribution of the two and four factors of the SPSPHD (*n* = 435).

Factor	Grand Mean (SD)	Item	Mean (SD)
Scale of Perceptions of Hemodialysis			
Factor 1: The atmosphere during hemodialysis	5.21 (6.05)	5	1.04 (1.21)
Factor 2: The negative emotions during hemodialysis	10.14 (2.56)	3	3.38 (0.33)
Scale of Self-Participation in Hemodialysis			
Factor 1: Creating a new life	18.72 (7.27)	7	2.67 (1.04)
Factor 2: Implementation of self-care	27.00 (7.37)	9	3.00 (0.82)
Factor 3: Adjusting and facing hemodialysis life	24.64 (7.70)	9	2.74 (0.86)
Factor 4: Active sharing and shared decisions	10.53 (4.16)	4	2.63 (1.04)

## Data Availability

The data used in this study are not available to the public due to ethical considerations.

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
