# Peer review of "The Scale of Perceptions and Self-Participation in Hemodialysis: Development and Psychometric Evaluation"

_healthcare, 2023, doi:10.3390/healthcare11233072_

Round 1

Reviewer 1 Report

Comments and Suggestions for Authors

This study presents the creation and validation of the "Scale of Perceptions of and Self-Participation in Hemodialysis" (SPSPHD), a tool designed to assess the self-participation of hemodialysis patients. The manuscript is well-written and elaborated.  

The abstract presents a study focused on developing and evaluating a "Self-Participation Dialysis Life Scale" for hemodialysis patients. It outlines the methodology used, including qualitative interviews, expert validation, and data analysis. The results indicate good reliability and validity of the scale, with two distinct parts related to patients' perceptions and self-participation in hemodialysis. While the study addresses an important aspect of patient care, it doesn't mention potential limitations or practical implications of the scale's use, which would be valuable for healthcare professionals and researchers seeking to apply or build upon this work.

The introduction of the manuscript highlights the significance of hemodialysis in Taiwan, particularly due to its high prevalence and cost. It emphasizes the challenges faced by end-stage renal failure patients undergoing hemodialysis, including a range of uncomfortable symptoms and psychological issues. The importance of patient-centered care and self-participation is stressed, along with the need for healthcare support that goes beyond traditional care experiences. However, there are a few shortcomings in this introduction. It doesn't clearly state the specific objectives of the study or the research questions being addressed. Additionally, while it mentions the importance of constructing the "Scale of Perceptions of and Self-Participation in Hemodialysis," it lacks a concise explanation of the methodology that will be used to develop this scale, leaving the reader somewhat unclear about the study's focus. Providing a more explicit research aim and a brief overview of the study design would enhance the clarity of this introduction.

The methodology section provides a comprehensive overview of the study's design, participants, instruments, procedure, ethical considerations, and data analysis. The use of psychometric testing and a structural questionnaire for a cross-sectional survey is well-defined. The inclusion and exclusion criteria for participants are clear, ensuring a focused and appropriate sample. The description of instruments, particularly the Hemodialysis Self-Management Instrument (HD-SMI) and the Scale of Perceptions and Self-Participation in Hemodialysis (SPSPHD), is detailed, although there is limited information on how the 42-item SPSPHD scale was constructed and its content validity. The procedure is also well-documented, including the process of developing the SPSPHD scale based on qualitative data.

In the results section, while the factors extracted through factor analysis are discussed, the interpretability and implications of these factors are not clearly elucidated. The practical significance and meaning of the identified factors should be addressed to help readers better understand their relevance. Second, the presentation of descriptive statistics could be more reader-friendly. Some of the mean values and standard deviations are reported in a somewhat fragmented manner, making it less accessible for readers to quickly grasp the scale's overall performance.

The discussion could benefit from a more critical analysis of the limitations of the study. The author acknowledges some limitations, such as the sample size and regional focus, but a more in-depth exploration of these limitations and their potential impact on the generalizability of the findings would enhance the discussion. Second, it would be valuable to discuss the practical implications of the study's findings. How might the SPSPHD be used in clinical practice or research, and what potential benefits could it offer to healthcare professionals and hemodialysis patients? Discussing the real-world applications of the scale could make the findings more meaningful. Finally, the author might consider addressing potential future research directions. Are there additional aspects of hemodialysis patients' experiences that the SPSPHD or other measures could explore? Are there opportunities for intervention or improvement in patient care based on the findings of this study?

The conclusion could be further improved by addressing a few points. It would be beneficial to reiterate the main practical implications of the study and offer specific recommendations for healthcare professionals based on the findings. How can the insights gained from this research be applied in clinical settings to enhance the care and well-being of hemodialysis patients?

Author Response

Point 1: Abstract: While the study addresses an important aspect of patient care, it doesn't mention potential limitations or practical implications of the scale's use, which would be valuable for healthcare professionals and researchers seeking to apply or build upon this work.

Response 1: The abstract has been revised (p.1, Line 29-31).

Point 2: Introduction: However, there are a few shortcomings in this introduction. It doesn't clearly state the specific objectives of the study or the research questions being addressed. Additionally, while it mentions the importance of constructing the "Scale of Perceptions of and Self-Participation in Hemodialysis," it lacks a concise explanation of the methodology that will be used to develop this scale, leaving the reader somewhat unclear about the study's focus. Providing a more explicit research aim and a brief overview of the study design would enhance the clarity of this introduction.

Response 2: The research aim has been revised on the 3rd paragraph of the introduction (p.2, Line 74-76). The methodology of developing this scale has revised on the last section of the instrument (p.3, Line 124-132).

Point 3: Methodology: The description of instruments, particularly the Hemodialysis Self-Management Instrument (HD-SMI) and the Scale of Perceptions and Self-Participation in Hemodialysis (SPSPHD), is detailed, although there is limited information on how the 42-item SPSPHD scale was constructed and its content validity. The procedure is also well-documented, including the process of developing the SPSPHD scale based on qualitative data.

Response 3: The information on constructing the 42-item SPSPHD (p.3, Line 124-126) and its content validity has been revised (p.3, Line 127-128).

Point 4: In the results section, while the factors extracted through factor analysis are discussed, the interpretability and implications of these factors are not clearly elucidated. The practical significance and meaning of the identified factors should be addressed to help readers better understand their relevance. Second, the presentation of descriptive statistics could be more reader-friendly. Some of the mean values and standard deviations are reported in a somewhat fragmented manner, making it less accessible for readers to quickly grasp the scale's overall performance.

Response 4: The section of the discussion has been revised (p.11, Line 316-338).

Point 5: The discussion could benefit from a more critical analysis of the limitations of the study. The author acknowledges some limitations, such as the sample size and regional focus, but a more in-depth exploration of these limitations and their potential impact on the generalizability of the findings would enhance the discussion. Second, it would be valuable to discuss the practical implications of the study's findings. How might the SPSPHD be used in clinical practice or research, and what potential benefits could it offer to healthcare professionals and hemodialysis patients? Discussing the real-world applications of the scale could make the findings more meaningful. Finally, the author might consider addressing potential future research directions. Are there additional aspects of hemodialysis patients' experiences that the SPSPHD or other measures could explore? Are there opportunities for intervention or improvement in patient care based on the findings of this study?

Response 5: The section of the discussion has been revised (pp.12-13, Line 399-425).

Point 6: The conclusion could be further improved by addressing a few points. It would be beneficial to reiterate the main practical implications of the study and offer specific recommendations for healthcare professionals based on the findings. How can the insights gained from this research be applied in clinical settings to enhance the care and well-being of hemodialysis patients?

Response 6: The conclusion has been revised (pp.13-14, Line 444-461).

Reviewer 2 Report

Comments and Suggestions for Authors

We read this article carefully and give the following personal opinion and recommendations:

1.In the abstract section, Line 13, "There is a lack of Self Participation Dialysis Life Scale for hemodialysis patients," there may be a lack of articulation with the preceding.

2.Are there formatting errors in the presentation of numbers in Line20,95,96,101,102,105,315?

3.The accuracy of the data in Line322, "Creating a New Life" (the Cronbach's α is 0.929) needs to be further verified, because "Creating a New Life" in Table 3 presents the Cronbach's α is 0.927.

4.In the discussion part of the article, this paper presents more of the specific results of the study, the discussion part is too simple.At the same time the data is a bit too much, which is not conducive to the reader's reading and understanding, and I hope that it can be presented in a more concise and logical order.

Comments on the Quality of English Language

The English in this paper is generally coherent and accurate, but there are some minor shortcomings, for example, in Line 15, is it necessary to use "hemodialysis patients" instead of 'hemodialysis patient"?

Author Response

Point 1: In the abstract section, Line 13, "There is a lack of Self Participation Dialysis Life Scale for hemodialysis patients," there may be a lack of articulation with the preceding.

Response 1: Two sentences have been added to the abstract (p.1, Line 13-15).

Point 2: Are there formatting errors in the presentation of numbers in Line20,95,96,101,102,105,315?

Response 2: Formatting errors in the text have been changed (p.1, Line 22; p.3 Line 108, 109, 115-117; p.12, Line 357-358).

Point 3: The accuracy of the data in Line322, "Creating a New Life" (the Cronbach's α is 0.929) needs to be further verified, because "Creating a New Life" in Table 3 presents the Cronbach's α is 0.957.

Response 3: The Cronbach's α of Creating a New Life has changed as 0.927 (p.9, Line 267).

Point 4: In the discussion part of the article, this paper presents more of the specific results of the study, the discussion part is too simple. At the same time, the data is a bit too much, which is not conducive to the reader's reading and understanding, and I hope that it can be presented in a more concise and logical order.

Response 4: We have reviewed and revised the section of results and discussion again (pp.5-13, Line 188-426).

Point 5: The English in this paper is generally coherent and accurate, but there are some minor shortcomings, for example, in Line 15, is it necessary to use "hemodialysis patients" instead of 'hemodialysis patient"?

Response 5: Hemodialysis patient has changed as hemodialysis patients (p.1, Line 17). The manuscript has reviewed by the professional English editor again.

Reviewer 3 Report

Comments and Suggestions for Authors

Thank you for the opportunity to review the manuscript “The Scale of Perceptions and Self-Participation in Hemodialysis: Development and Psychometric Evaluation” (healthcare 2642535).

The authors develop and evaluate a self-participation dialysis life scale for hemodialysis patient. The items for the self-participation dialysis life scale were confirmed through qualitative interviews.

Line 65: Please define or add a reference (ideally both) to explain what you mean by ‘patient centred care’. It means different things to different people. 

The introduction and theoretical basis must be better connected with discussion and conclusion.

The relevant literature must be discussed with the results. Furthermore, the work would benefit from a table or graph summarizing all relevant results.

Objectives and the rationale of the study clearly stated. The authors also emphasized the strengths and limitations of their study.

However please check language, please.

Comments on the Quality of English Language

must carefully improved

Author Response

Point 1: Line 65: Please define or add a reference (ideally both) to explain what you mean by ‘patient centred care’. It means different things to different people.

Response 1: The definition and a citation have been added to the last paragraph of the introduction (p.2, Line 62-69).

Point 2: The introduction and theoretical basis must be better connected with discussion and conclusion.

Response 2: The discussion and conclusion have been revised (pp.12-13, Line 399-425).

Point 3: The relevant literature must be discussed with the results. Furthermore, the work would benefit from a table or graph summarizing all relevant results.

Response 3: The section of discussion has been revised (pp.12-13, Line 399-425). The results already has 5 tables and 3 figures for the findings so that we cannot add a table or graph summarizing all relevant results.

Point 4: However please check language, please.

Response 4: The manuscript has reviewed by the professional English editor again.

Round 2

Reviewer 3 Report

Comments and Suggestions for Authors

The authors improved the paper according to the reviewers suggestions.

Comments on the Quality of English Language

is now better

Author Response

Point 1: The authors improved the paper according to the reviewers suggestions.

Response 1: Thanks so much.

Point 2: The Quality of English Language is now better.

Response 2: Thanks a lot.

Point 3: Does the introduction provide sufficient background and include all relevant references? Can be improved.

Response 3: We have reviewed and revised the text again (pp.1-15).

Point 4: Are all the cited references relevant to the research? Can be improved.

Response 4: We have reviewed and revised the text again (pp.1-15).